# What dictates soft clay-like lithium super-ionic conductor formation from rigid salts mixture

Sunny Gupta [1,2], Xiaochen Yang[1,2] & Gerbrand Ceder [1,2]

Soft clay-like Li-superionic conductors, integral to realizing all-solid-state batteries, have been recently synthesized by mixing rigid-salts. Here, through computational and experimental analysis, we clarify how a soft clay-like material can be created from a mixture of rigid-salts. Using molecular dynamics simulations with a deep learning-based interatomic potential energy model, we uncover the microscopic features responsible for soft clay-formation from ionic solid mixtures. We find that salt mixtures capable of forming molecular solid units on anion exchange, along with the slow kinetics of such reactions, are key to soft-clay formation. Molecular solid units serve as sites for shear transformation zones, and their inherent softness enables plasticity at low stress. Extended X-ray absorption fine structure spectroscopy confirms the formation of molecular solid units. A general strategy for creating soft clay-like materials from ionic solid mixtures is formulated.

Mechanically soft materials that can be easily deformed by hand are ubiquitous in nature, such as natural soft-clay[1], dough[2], gel[3], etc. Such materials usually consist of hard and soft components[1,3,4], where the former gives rigidity, while the latter gives soft rheological behavior. For example, in natural soft-clays, hard components are layered-minerals such as pyrophyllite[1], while the soft component is water. Soft materials are highly sought after in various applications such as in flexible electronics[5], pharmaceuticals for drug delivery[4], and all-solid-state batteries to improve fabrication, and interfacial kinetics[6–8].

Recently, such soft clay-like materials were introduced into the energy storage field as potential fast Li-ion conductors. Intriguingly, a soft clay-like material was synthesized by ball milling two rigid salts, $LiCl$ and $GaF_3$, at room temperature (RT) without water[9,10]. The resulting amorphous solid had a high RT Li-ion conductivity of ~4 mS/cm, which is comparable to that of liquid organic electrolytes. Pliable solid-state ionic conductors are of particular interest as they have the potential to facilitate the fabrication of solid-state batteries and accommodate the swelling of active cathode materials[11]. The fact that a soft material can be created by merely mixing two rigid ionic solids contradicts conventional thinking. Understanding the physical mechanism behind mechanical softness in such materials and

identifying the general criteria for soft-clay formation from mixtures of ionic solids could help to establish design principles for creating innovative soft materials. Combining the exotic properties that can be achieved in ionic materials, such as magnetism[12,13], polaronic optical absorption[14], color centers host[15], magneto-optical response[16], etc., and the concept of creating mechanically soft systems from mixtures of ionic solids has far-reaching implications.

This study presents a combined computational and experimental analysis of the microscopic mechanism governing soft-clay formation upon mixing two rigid salts, $GaF_3$ and $LiCl$. By training a deep learning-based interatomic potential energy (PE) model, we were able to explore the mechanical behavior of an amorphous structure with ~10k atoms using molecular dynamics (MD) simulations. The results reveal that the formation of molecular solid-like (MS) units from the chemical reaction between the hard solids, is responsible for the material's mechanical softness. These units serve as sites for shear transformation zones and are inherently soft, enabling plasticity at low stress. Extended X-ray absorption fine structure (EXAFS) of the synthesized soft clay-like material confirmed the formation of these molecular units. This study provides a detailed understanding of the microscopic mechanism of soft-clay formation from rigid ionic solid mixtures and

[1]Department of Materials Science & Engineering, University of California Berkeley, Berkeley, CA 94720, USA. [2]Materials Sciences Division, Lawrence Berkeley National Laboratory, Berkeley, CA 94720, USA. ✉e-mail: gceder@berkeley.edu

proposes criteria and design strategies for realizing other mechanically soft materials.

## Results

LiCl and $GaF_3$, two rigid salts with melting points 878 K and 1070 K, respectively, were recently found[9] to form a soft clay-like amorphous solid with a glass transition at $T_g$ ~ −60 °C, when ball milled in a molar ratio $x$LiCl:$GaF_3$ ($2 \leq x \leq 4$) for 18 h at RT (a powder state remained for $x < 2$). To explore the reaction mechanism and mechanical behavior, a deep learning-based interatomic PE model was trained for the Ga-F-Li-Cl chemical system using DeepMD[17,18] (details in Methods and Supplementary Note 1). Training was performed using the atomic forces and energies of ab initio molecular dynamics (AIMD) trajectories of 13 stable crystalline phases in the Ga-F-Li-Cl quaternary chemical space, and 3 slab-like structures of LiCl|$GaF_3$ (where multilayer slabs of [001] LiCl and [001] $GaF_3$ having 3 different thicknesses are interfaced "|") (see Supplementary Fig. 1 and Supplementary Table 1 for the full list of structures). The atomic configurations were obtained by melting and quenching each structure using AIMD simulations. In total, >700k AIMD frames were generated, from which 250k atomic configurations were randomly chosen for training. Our trained model was validated on a subset of training structures, and the RMSE error in energy and forces were <1 meV/atom and <70 meV/Å after training (Supplementary Fig. 2). The trained model was also tested on atomic forces and energies of several additional structural configurations (not included in the training and validation set), and benchmarked against their bulk modulus calculated with density functional theory (DFT), and tensile stress-strain response and Li-ion conductivity calculated with DFT-AIMD, and showed good agreement with actual DFT values (see Supplementary Table 2, and Supplementary Figs. 3, 4). We also examined our trained PE model's accuracy in describing the properties of several amorphous systems in the Ga-F-Li-Cl chemical space and found good agreement between model's prediction and DFT values (see Supplementary Note 1.3 and Supplementary Fig. 5).

In recent experiments[9,10], the amorphous clay-like solid obtained from ball-milled LiCl and $GaF_3$ was found to contain domains of unreacted parent constituents distributed randomly within the amorphous matrix. To emulate conditions observed in experiments, where LiCl and $GaF_3$ particles come into contact and react during ball milling, a specific procedure was designated to create large supercells with domains of each constituent (LiCl and $GaF_3$). The procedure involved firstly creating a slab-like geometry of LiCl|$GaF_3$, where a multilayer slab of [001] LiCl and [001] $GaF_3$ was interfaced in a LiCl:$GaF_3$ molar ratio of 2. This slab-like structure was subjected to an AIMD simulation

under the NVT ensemble for $t = 5$ ps at $T = 800$ K to equilibrate the interface, and then relaxed at $T = 0$ K to a local energy minimum. The resulting structure, as shown in Supplementary Fig. 6a, was then periodically repeated along the two directions perpendicular to the interface, and a ~2.5 nm cubic particle was cut out from it. Eight of these identical ~2.5 nm aperiodic particles, adding up to ~10k atoms, were arranged in a larger cell of size ~5.4 nm (Supplementary Fig. 6b). To achieve many distinct interfacial environments, each of the ~2.5 nm particles was rotated 90° with respect to their neighbors, such that the axis normal to the interface between LiCl and $GaF_3$ in each particle did not align in the same direction as that of the neighboring particles (additional details in Supplementary Note 2 and Supplementary Fig. 6). Relaxing the cell in LAMMPS[19] using the trained deep learning-based PE model at $T = 0$ K, densified the system by removing the vacuum between each ~2.5 nm cube of material. This relaxed structure with ~10k atoms was subjected to a high-temperature classical MD simulation (hereafter, unless specified, all MD simulations were done with LAMMPS[19] using the trained deep learning-based PE model) under the NVT ensemble for $t = 50$ ps at $T = 900$ K (see Supplementary Note 2.1 for the system's behavior at other temperatures) to allow mixing of the atoms. To obtain a representative structure in which domains of reactant materials are visible, an atomic configuration at $t = 50$ ps was selected and this configuration was relaxed in LAMMPS at $T = 0$ K to a local energy minimum (this procedure is analogous to the commonly used melt and quench techniques[20,21] to create amorphous structures). The relaxed structure was further equilibrated in an MD simulation under the NPT ensemble for a duration of 2 ns at $T = 300$ K and zero external stress, until there was no further change in density, lattice constant, angles, or total PE. The resulting amorphous structure is shown in Fig. 1a and exhibits chemical heterogeneity with domains of unreacted LiCl and $GaF_3$, as well as the formation of $GaCl_3$-like units which are molecular complexes that have a 4-fold Ga-Cl tetrahedral coordination similar to that in bulk $GaCl_3$ (Fig. 3b). The specific volume of the structure was computed at different temperatures, and a glass-like transition was observed at $T_g$ ~ −58 °C (Supplementary Fig. 9), in good agreement with experiments[9]. Fig. 1b shows the element-wise radial pair distribution function $g(r)$ before and after the high-temperature ($T = 900$ K) MD simulation of $t = 50$ ps. The Ga-F and Li-Cl peak decrease, while the Ga-Cl and Li-F peak increase, indicating that anion exchange occurs when the two salts are mixed. This anion exchange is consistent with thermodynamic driving force. Considering all possible phases that can form when LiCl and $GaF_3$ react, using entries from the Materials Project database[22], the thermodynamic driving force is found to be maximum (reaction energy is

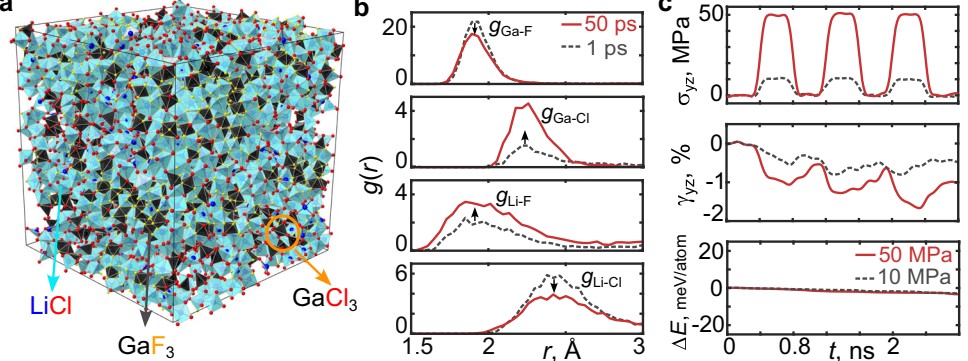

**Fig. 1 | A representative amorphous structure displaying the occurrence of anion exchange and exhibiting a soft mechanical behavior. a** An amorphous Ga-F-Li-Cl structure constructed for computational modeling, containing domains of LiCl, $GaF_3$, and $GaCl_3$-like molecules formed during the high-temperature MD simulations. **b** The element-wise radial pair distribution function $g(r)$ plotted at $t = 1$ ps and $t = 50$ ps of the MD simulation at $T = 900$ K. **c** The applied external shear stress $\sigma_{yz}$, accumulated shear strain $\gamma_{yz}$, and total potential energy change $\Delta E$ of the amorphous structure as a function of time in the MD simulation at $T = 300$ K. The external shear stress ranged from 10 to 50 MPa. The accumulated shear strain after three cycles is non-negligible ($\gamma_{yz} \neq 0$), signifying permanent deformation.

most negative) for the anion exchange reaction $3LiCl + 2GaF_3 \rightarrow 1Li_3GaF_6 + 1GaCl_3$, $E_{rxn.} = -93$ meV/atom (Supplementary Fig. 8).

To determine whether our amorphous structure (Fig. 1a) exhibits soft mechanical behavior similar to that observed in experiments, we simulated the response of the system in a stress-controlled MD simulation under an NPT ensemble at 300 K. The simulation involved subjecting the system to an external shear stress, specifically setting the $yz$-component $\sigma_{yz}$ to either 10 MPa or 50 MPa, while keeping all the other external stress components at zero. The shear stress was applied for 400 ps and then released for 400 ps, and this stress pulse was repeated three times. Fig. 1c shows the resulting shear strain $\gamma_{yz}$, and total PE $\Delta E$. The external stresses' $yz$-component was arbitrarily chosen and was found not to influence the results (see Supplementary Fig. 12 for results when $xz$-component of external stress was applied and other components were kept at zero). The accumulated shear strain after three loading cycles is non-negligible, and the structure exhibits permanent deformation and plastic behavior, even at low $\sigma_{yz} = 10$ MPa. This shows that the amorphous structure is mechanically soft, similar to what is observed in experiments. The change in PE after deformation is small <3 meV/atom. The small change in PE upon application of stress cycles is expected in glasses, which typically have a fractal-like PE surface[23,24], and indicates that the amorphous structure (Fig. 1a) is in a meta-basin. A glassy material can hop between near-degenerate sub-basins within a meta-basin and achieve plasticity even at low stresses[24]. Analysis at even lower shear stresses $\sigma_{yz} < 10$ MPa, could not be performed due to stress fluctuations of that order in the simulations.

To further investigate the microscopic features responsible for the material's soft-plastic response, we performed another MD simulation at a constant strain rate and $T = 300$ K (additional details in Methods). Such strain-controlled MD simulations are commonly employed[24,25] to identify the microscopic plastic events, as they enable the separation of the elastic and plastic regions based on strain magnitudes. Fig. 2a shows the shear stress-strain response of the structure (Fig. 1a) at an applied shear strain rate of $10^9$ s$^{-1}$ on the $yz$ plane. While the obtained yield stress under this very high strain rate is ~0.5 GPa (Fig. 2a), the stress is expected to be much lower under a lower strain rate[24,25]. Lower strain rates, as used in experiments are not directly accessible in molecular dynamics, and methods such as metadynamics[24] would have

to be used. However, our results using a constant stress (Fig. 1c) demonstrate that the material is indeed soft. A non-elastic plastic response is evident at $\gamma_{yz} > 0.05$ in Fig. 2a. Deviation from the elastic response in amorphous materials can be quantified by the non-affine displacements ($D^2_{min}$)[24,26]. A method previously[26] reported to calculate $D^2_{min}$ was used here (see details in Supplementary Note 5). Fig. 2b-d shows snapshots of the structures at three different (i–iii) $\gamma_{yz}$ values with the local value of $D^2_{min}$ represented by the color legend. In structure i (Fig. 2b), at a low shear strain, the areas with non-zero $D^2_{min}$ indicate local plastic deformation, commonly known as Shear Transformation Zones (STZs)[24,26,27] in amorphous materials. In structure ii (Fig. 2c), where $\gamma_{yz}$ has increased, these STZs form shear bands[24–27]. Moreover, in structure iii (Fig. 2d), with further increased $\gamma_{yz}$, localized STZs similar to those observed in structure i (Fig. 2b) reappear. The MD simulation at a constant strain rate (Fig. 2a–d) reveals that our structure exhibits plastic deformation by forming STZs and shear bands, which are characteristic of plastic deformation in amorphous materials[24–27].

To understand the local chemical environment of STZs responsible for plastic deformation, we plotted in Fig. 2f the radial distribution function $g(r)$ around Ga atoms with the highest and lowest $D^2_{min}$ values (see Supplementary Fig. 15 for a similar plot of $g(r)$ around Li atoms). The $g(r)$ was averaged over the strain interval $\gamma_{yz} = 0.09$–0.14. We found that Ga atoms in the areas involved in plastic deformation have a Cl-rich environment, as evidenced by the lower Ga-F peak for Ga atoms with $D^2_{min} > 90\%$ of maximum compared to those with $D^2_{min}$ values <10% of maximum (Fig. 2f). Additionally, we visually examined the local structure in STZs and found that it mostly consists of GaCl$_3$-like and Cl-rich GaCl$_x$F$_y$ complexes (Fig. 2e). As illustrated in Fig. 3b bulk GaCl$_3$ is a molecular solid consisting of Ga$_2$Cl$_6$ complexes that are Van der Waals bonded to each other. The weak intermolecular interactions in molecular solids render them intrinsically very soft[28]. The nature of this weak bonding rationalizes why strain localizes in areas where Ga is mostly coordinated by Cl in our simulations. These findings indicate that the soft-plastic behavior in the 2LiCl-1GaF$_3$ structure arises due to the formation of molecular solid-like GaCl$_3$-units and Cl-rich GaCl$_x$F$_y$ complexes during anion exchange. These complexes are activated at low stresses to form STZs, ultimately leading to soft-plastic deformation. Thus, the formation of molecular solid units is the key to the soft clay-like plastic response.

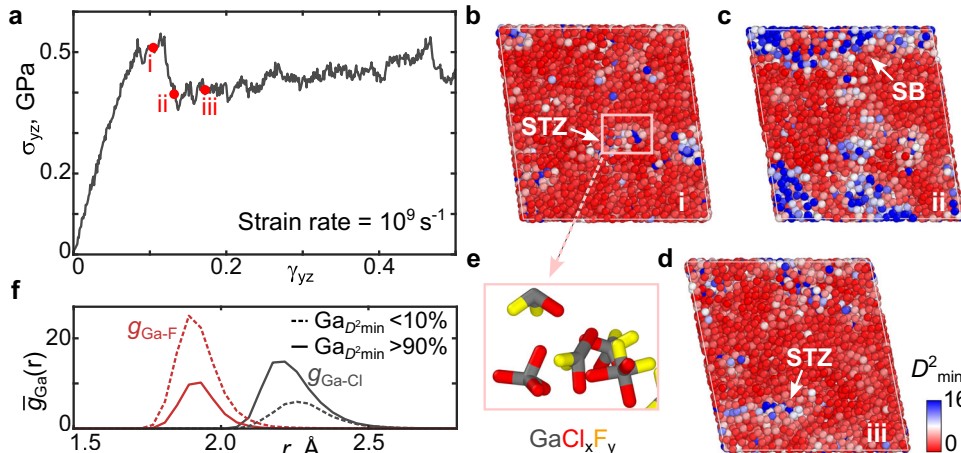

**Fig. 2 | Microscopic features responsible for the material's plastic response.** **a** Shear stress ($\sigma_{yz}$) strain ($\gamma_{yz}$) response of the amorphous structure at $T = 300$ K at a constant applied shear strain rate of $10^9$ s$^{-1}$. **b–d** Shows snapshots of the material at different strain values i–iii in **a**. The color (red-low and blue-high) represents the value of non-affine displacement $D^2_{min}$, at an interval of $\Delta\gamma_{yz} = 0.01$. **b**, i the shear transformation zone (STZ) formed is indicated by an arrow and marked by a pink square. **c**, ii the arrow points to shear bands (SB). **e** Shows the atoms that are part of

the STZ in the pink square in **b**, i and have large $D^2_{min}$ values. These are mostly GaCl$_3$-like and Cl-rich GaCl$_x$F$_y$ complexes. **f** The Ga-Cl and Ga-F pair distribution function $g(r)$ of Ga-atoms with largest ($D^2_{min} > 90\%$ of maximum) and smallest ($D^2_{min} < 10\%$ of maximum) non-affine displacement values. The $g(r)$ values were averaged over the strain interval $\gamma_{yz} = 0.09$–0.14.

To validate anion exchange and the formation of molecular units in the $2LiCl-1GaF_3$ and $3LiCl-1GaF_3$ soft clay-like materials, we prepared samples by mechanochemically mixing (ball milling) LiCl and $GaF_3$ precursors in the ratio of 2:1 and 3:1, as previously reported[9] (further experimental details in Methods). The inset in Fig. 3a shows an image of the resulting soft clay-like material bent into a specific shape. The RT Li-ion conductivity measured using electrochemical impedance spectroscopy (EIS) was found to be ~2.9 mS/cm (details in Methods, Supplementary Fig. 11), similar to that reported in ref. 9 To understand the local atomic chemical environment of Ga atoms in the soft clay-like materials we performed EXAFS[29]. Fig. 3a compares the GaK-edge EXAFS of the ball-milled $2LiCl-GaF_3$ and $3LiCl-GaF_3$ with that of bulk $GaF_3$ and $GaCl_3$. The Ga EXAFS curves of $2LiCl-1GaF_3$ and $3LiCl-1GaF_3$ lie between the curves of bulk-$GaF_3$ and bulk-$GaCl_3$ indicating contributions from both $GaF_3$-like as well $GaCl_3$-like units. This confirms that anion exchange takes place during the ball milling of a $LiCl-GaF_3$ mixture forming $GaCl_3$-like molecular units, consistent with the observations in the MD simulations (Fig. 1b). The weak magnitude of Fourier transformed EXAFS at higher radial distance indicates a highly disordered Ga environment, which supports the amorphous nature of the soft clay-like samples (see Supplementary Fig. 13 for the EXAFS plotted to a radial distance of 6 Å). XRD of the soft clay-like materials

(Supplementary Fig. 10e) did not reveal any peaks corresponding to bulk $GaCl_3$, indicating that no distinct crystalline $GaCl_3$ phase formed.

To achieve soft clay-like mechanical behavior, it is necessary to have both soft and hard components and them being interfaced with each other[1]. In our particular case, hard components are provided by unreacted LiCl and $GaF_3$, while the $GaCl_3$-like molecular component provides softness. Soft clay will not form if the soft component is deficient or in excess, analogous to the appropriate ratio[30] of water and pyrophyllite-like minerals to form natural soft-clay. If the soft component is in excess, soft-units can phase segregate into a macroscopic phase, which is detrimental for achieving a soft-clay-like property. It has been seen that $GaF_3$-rich compositions, $x < 2$ in $xLiCl:GaF_3$, do not form a soft-clay[9,31] indicating that the right amount of soft units is required to achieve soft clay-like mechanical behavior. Moreover, if both the soft and hard components phase segregates into macroscopic separate phases, then soft clay-like deformation will also not be achieved. Indeed, ball milling the compounds that would constitute the terminal products of anion exchange, LiF and $GaCl_3$, does not lead to a soft-clay. This points at the importance of the kinetics of anion exchange, in addition to the mixture having the right ratio of components. Anion exchange must occur but should not be completed, and the products should not be separated into macroscopic phases.

In an attempt to find other soft-clay-forming mixtures, we searched the Materials Project database[22] and identified three other molecular solids, $SbCl_3$, $InI_3$, and $GaI_3$, which can form through anion exchange from rigid solid mixtures $1SbF_3-3LiCl$, $1InBr_3-3LiI$, and $1GaF_3-3LiI$, respectively (Supplementary Note 3). The crystal structures of $SbCl_3$ and $InI_3$, depicted in Fig. 3c, d, consist of molecular units that are held together through Van der Waals interactions, while $GaI_3$ has a similar crystal structure as $GaCl_3$ (Fig. 3b). Table 1 displays the different salt combinations evaluated in this study for possible soft-clay formation. The mixtures were ball-milled and characterized by XRD to identify any crystalline phases formed. We find that in the case of $3LiCl-1SbF_3$, $3LiI-1InBr_3$, and $3LiI-1GaF_3$, a powder state remains after ball milling and a soft-clay does not form (hereafter called non-clay). Additionally, in all three non-clay cases, peaks corresponding to crystalline molecular solid phases ($SbCl_3$, $InI_3$, and $GaI_3$) were visible in XRD (see Table 1, Supplementary Fig. 10). In contrast, for $2LiCl-1GaF_3$ and $3LiCl-1GaF_3$, a mechanically soft solid was obtained after ball milling (hereafter called clay). XRD spectra of the $3LiCl-1GaF_3$ (clay) only showed peaks corresponding to LiCl (unreacted precursor), while no peaks corresponding to any crystalline $GaCl_3$ were visible (see Table 1, Supplementary Fig. 10). The appearance of peaks corresponding to crystalline molecular solid phases in the XRD of the non-clay systems, signifies that the molecular solid units have separated into macroscopic phases. Moreover, in the non-clay systems $3LiCl-1SbF_3$, and $3LiI-1InBr_3$, XRD peaks corresponding to the anion exchanged products (apart from MS), LiF and LiBr, were also observed, signifying that the anion exchange products have separated into macroscopic phases. These findings reveal that non-clay formation is linked with the separation of anion-exchanged products into macroscopic phases, thereby validating the role of kinetics in soft-clay formation. To form a

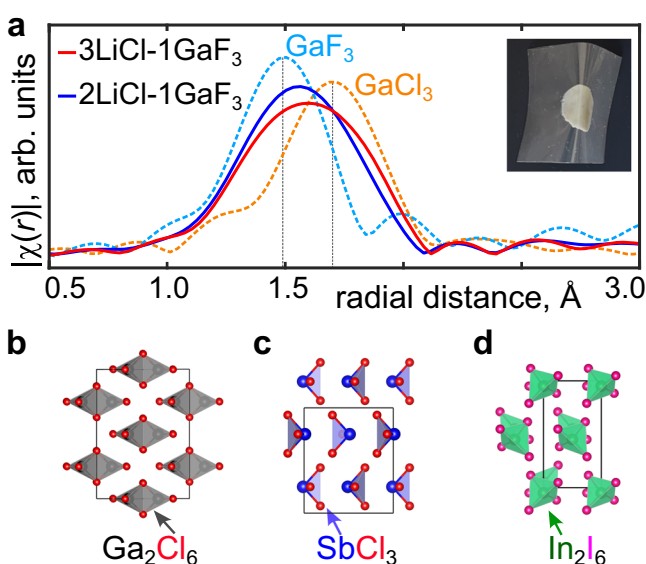

**Fig. 3 | Experimental confirmation of molecular solid-like units formation. a** Ga K-edge EXAFS of different materials, bulk $GaF_3$, bulk $GaCl_3$, and clay-like $2LiCl-1GaF_3$ and $3LiCl-1GaF_3$. $|\chi(r)|$, magnitude of Fourier transformed EXAFS. Inset shows an image of synthesized soft clay-like $2LiCl-GaF_3$ bent into a particular shape. **b–d** Bulk crystal structure of molecular solids $GaCl_3$, $SbCl_3$, and $InI_3$, respectively. The Ga, Cl, Sb, In, and I atoms are shown in black, red, blue, green, and pink color, respectively.

**Table 1 | Showing different salt combinations (precursors) considered in this study for possible soft-clay formation**

| Precursors | Predicted products with maximum $E_{rxn}$ | Possible molecular solid component | XRD peak after BM | Rheological behavior |
|---|---|---|---|---|
| $2LiCl + 1GaF_3$ and $3LiCl + 1GaF_3$ | $Li_3GaF_6 + GaCl_3$ | $GaCl_3$ | LiCl | soft clay-like |
| $3LiCl + 1SbF_3$ | $LiF + SbCl_3$ | $SbCl_3$ | $LiF + SbCl_3$ | powder |
| $3LiI + 1InBr_3$ | $LiBr + InI_3$ | $InI_3$ | $LiBr + InI_3$ | powder |
| $3LiI + 1GaF_3$ | $LiF + GaI_3$ | $GaI_3$ | $GaI_3$ | powder |

The predicted products are based on the maximum thermodynamic driving force with the maximum (most negative) reaction energy ($E_{rxn}$). Possible molecular solid component that can form is also listed. Additionally, the crystalline phases in the product, formed after ball milling (BM) the precursors in experiments are identified with X-ray diffraction (XRD) and listed under the column—XRD peak after BM. The rheological behavior of ball-milled products is also tabulated.

soft-clay, anion exchange must occur, but the products should not phase segregate into macroscopic separate phases. Furthermore, we tracked the XRD peaks of phases formed at different instances of time during ball milling (see Supplementary Fig. 10d). For a representative non-clay material $3LiI·1InBr_3$, XRD peaks corresponding to LiBr and $InI_3$ were seen after just 20 min of ball milling, indicating a rapid anion exchange and phase separation, while for the soft clay-like $3LiCl·1GaF_3$, no signatures of anion exchange completion or phase separation were seen even after 24 h of ball milling. This indicates that the inherent chemistry of $GaF_3$ impedes kinetics of complete anion exchange and phase separation. Additionally, in non-clay systems, strategies to suppress kinetics, such as regulating temperature during ball milling, can be used for soft-clay formation.

## Discussion

Our combined computational and experimental study shows that when a rigid salt mixture of $2LiCl·1GaF_3$ mechanochemically reacts, the resulting system mechanically behaves like a soft-clay, as evidenced in both MD simulations (Fig. 1c) and experiments (Fig. 3a inset), consistent with prior studies[9,31]. We find that during the ball milling reaction, partial anion exchange takes place and $GaCl_3$-like molecular solid (MS) units are formed, which is consistent with the thermodynamic driving force (Supplementary Fig. 8). This process was confirmed both with MD simulations (Figs. 1a, b, 2b) and EXAFS (Fig. 3a). MD simulations of the shear stress-strain response of the material (Fig. 2) show that MS-like units serve as sites for shear transformation zones which get activated at low stress and lead to soft-plastic deformation. These findings point to the formation of molecular solid units as a result of the partial anion exchange, as the key to the soft clay-like mechanical response. Our experiments on other ion-exchange solids indicate that careful tuning of the composition and processing is required to obtain the proper soft mechanical response. Some salt mixtures have a thermodynamic driving force (TDF) for anion exchange reaction, but if the anion exchange reaction does not lead to MS formation, soft-clay will not be obtained. This phenomenon was observed in prior studies[9,31] for salt mixtures such as $1NaCl·1GaF_3$, $3LiCl·1InF_3$, $6LiCl·1Ga_2O_3$, $3LiOH·1GaF_3$, $1Li_2O·1GaF_3$, and $3LiCl·1LaF_3$, where soft-clay formation was unsuccessful. Except for the case of $3LiCl·1InF_3$, where there is no TDF for anion exchange reaction, in all other cases, the salt mixtures have a TDF for anion exchange reaction, but the anion exchange reaction does not lead to MS formation (see Supplementary Note 3). In contrast, the $3LiBr·1GaF_3$ mixture was found[9] to exhibit soft clay-like mechanical behavior driven by the favorable reaction energy of an anion exchange between LiBr and $GaF_3$ to form $GaBr_3$ (see Supplementary Note 3), which is a molecular solid and has a similar crystal structure as $GaCl_3$ (Fig. 3b). We searched the Materials Project database for other molecular solids and the anion exchange reaction with the highest TDF to form them (see Supplementary Table 3), which has the potential to form soft-clay. Additionally, in recent experiments[10], ball milling a mixture of $2LiCl·xAlF_3·(1-x)GaF_3$ revealed soft-clay-like behavior when $x - 0.8$. For $x > 0.8$ ($AlF_3$-rich), a powder state remained, while for $x < 0.8$, a viscous gel-like state was obtained. $AlF_3$ does not have any TDF to react with LiCl or $GaF_3$. This is likely the reason why $AlF_3$-rich compositions do not form a soft-clay, and why $GaF_3$ remains essential for obtaining soft-clay-like mechanical properties in the $2LiCl·xAlF_3·(1-x)GaF_3$ mixture. The authors found that for compositions $x < 0.8$, a viscous state was formed. We attribute this to the fact that addition of $AlF_3$ lowers the effective melting point of the salt mixtures[10], and the resulting system might be a mixture of rigid solids dispersed in a liquid with viscous-like behavior[3].

The kinetics of anion exchange is also critical for soft-clay formation. If the anion exchange reaction's kinetics is fast, such that it leads to anion exchange completion and phase segregation of the molecular species into macroscopic phases, then the MS-like soft units will not be interfaced between the hard solids, and soft clay-like

mechanical response will not occur. This phenomenon is observed in salts mixture $3LiCl·1SbF_3$, $3LiI·1InBr_3$, and $3LiI·1GaF_3$ (Table 1), where the XRD spectra (Supplementary Fig. 10) of the ball-milled products showed peaks corresponding to crystalline molecular solid phases, indicating that the MS units have separated into macroscopic phases. Suppressing the kinetics of anion exchange in such non-clay forming cases may be possible if the temperature during ball milling can be regulated. It is intuitive that the appropriate ratio of salts in the mixture is also crucial for soft-clay formation; neither of the salt components should be in much excess. Although important, the exact ratio of salts in the mixture to form soft-clay is challenging to predict, apriori. Our combined computational simulations and experimental results suggest that to form soft-clay from ionic solids, three criteria must be met - (i) the combination of salts must have a thermodynamic driving force for anion exchange leading to units that form molecular solids, (ii) the kinetics of the reaction must be slow enough to avoid complete anion exchange and phase separation, and (iii) the ratio of salts must be in an appropriate range, ensuring that neither of the salt components is in much excess. Until now, only the $LiCl·GaF_3$, and $LiCl·AlF_3·GaF_3$ systems have demonstrated soft-clay-like properties. The underlying chemistry of Ga-F-Cl chemical space appears to assist in satisfying all the three criteria for soft-clay formation, however, the microscopic reasons that distinguish Ga-F-Cl chemical systems warrant further investigation.

The mechanism behind mechanical softness in clay-like Li superionic conductors synthesized from a mixture of rigid salts, in this and prior studies[9,10,31], bears resemblance to both natural soft-clays[1,30,32], and synthetic soft-clays[33-35] formed from a blend of hard minerals and soft polymers. The unifying feature responsible for the pliability of all these soft-clays is the presence of soft components within a rigid matrix. In natural soft-clays, a mixture of hard minerals and water, plasticity arises when platelet-like hard minerals slide over each other upon the addition of water[30,32], which acts as the soft component. Moreover, synthetic soft-clays composed of hard minerals dispersed in a soft polymer matrix exhibit plasticity through shear banding of the soft polymers, accompanied by momentum transfer to the rigid minerals[33,34]. Analogously, in soft clay-like Li superionic conductors created from a mixture of rigid salts, we find that plasticity occurs through the formation of shear transformation zones (STZs) and shear bands at the sites of soft MS-like units between the hard solids (Fig. 2). The inherent softness of MS-like units leads to the activation of STZs at low stresses, resulting in a soft clay-like mechanical response. Additionally, in natural and synthetic soft-clays, factors such as the ratio of rigid to soft units, particle size distribution, specific surface area of particles, and temperature also contribute to the plasticity[32,33]. Although determining the precise role of these factors can be challenging, we anticipate that they will significantly influence the pliability of clay-like materials synthesized from a mixture of rigid salts. The soft clay-systems examined in this study are in an amorphous state and exhibit a glass transition[9] at $T_g \sim -60$ °C. This implies that at room temperature, there might be regions in the amorphous system that are in a supercooled liquid state[36,37] and those regions may contain molecular units that are critical to the system's soft mechanical response. Finding molecular solids capable of forming glass and remaining in a supercooled liquid state at room temperature, as well as developing methods to synthesize such molecular solid based compounds, can aid in the design of other soft-clay systems and requires further exploration.

In summary, we have uncovered the microscopic mechanism of soft-clay formation from the ionic solid mixtures by studying the mechanical behavior of amorphous $2LiCl·1GaF_3$ using MD simulations and with a deep learning-based interatomic PE model. We find that the formation of molecular solid-like units as a result of anion exchange is the key to soft clay-like mechanical behavior. MS-like units serve as the sites for STZs which get activated at low stress and lead to soft-plastic deformation. Additionally, our combined computational simulations

and experimental results suggest that to form soft clay-like systems from generic ionic solid mixtures, three criteria must be met - (i) the combination of salts must have a thermodynamic driving force for anion exchange leading to units that form molecular solids, (ii) the kinetics of the reaction must be slow enough to avoid complete anion exchange and phase separation, and (iii) the ratio of salts must be in an appropriate range, ensuring that neither of the salt components is in much excess. Until now only Li-based systems have shown pliable behavior, however, we believe that these strategies are not limited to Li-based systems and can be applied to discover other beyond-Li soft clay-like systems if mixtures with appropriate reaction energy can be found. Additionally, exploiting the inherent properties of ionic solids, these approaches can be used to create flexible magnets, electronic conductors, and other pliable superionic (Mg, Ca, etc.) conductors with wide-ranging implications.

## Experimental details

### Synthesis
Mechanochemical ball milling was used for all anion exchange reactions. LiCl (ACS Reagent 99%), $GaF_3$ (Thermo Invitrogen, 99.85%), LiI (Alfa Aesar, 99%), $SbF_3$ (Sigma Aldrich, 99.8%), and $InBr_3$ (Sigma Aldrich, 99.99%) were used as precursors. Stoichiometric amounts of the precursors were dispersed into Ar-filled zirconium oxide ball-mill jars and then planetary ball-milled (Retch PM 200) at a rate of 450 rpm for 24 h. In each jar, the total amount of precursors was ≈1 g. Five 10-mm (diameter) and ten 5-mm-(diameter) zirconia balls were used as the grinding media.

### Electrochemical characterization
The lithium-ion conductivity was evaluated using EIS with platinum foil as blocking electrodes at $T = 25\,°C$. The samples are assembled and sealed in a swagelok-type cell in an Ar-filled glovebox. EIS measurements were performed using an EC-Lab Electrochemistry SP300 system (Biologic). The measurements were conducted at the initial open-circuit voltage in the frequency range of 7 MHz –10 mHz with the application of a 10-mV signal amplitude.

### Structural characterization
XRD patterns of the as-prepared materials were collected using a Rigaku MiniFlex diffractometer with Cu Kα radiation.

### Extended X-ray absorption fine structure spectroscopy
Ga K-edge EXAFS was performed at beamline 7BM of NSLS-II, Brookhaven National Laboratory. The measurements were performed in transmission mode using a Si(111) monochromator. A Rhenium foil (10535 eV) was simultaneously measured with the experimental measurements to calibrate the energy of the individual datasets. To prevent air exposure, samples were sealed between polyimide tape. The EXAFS spectra of the Ga edge were calibrated and normalized using the Athena software packages[42]. The background contribution was limited below radial distance (Rbkg) = 1.0 using the built-in AUTOBK algorithm. The extracted EXAFS signal was weighted by $k^3$ to accentuate the high-energy oscillations and then Fourier transform using a Hanning window function to plot the spectra in $R$-space. Since the Fourier transform was not phase corrected, the $R$ values obtained from Fourier transform are shorter than the actual distances[43].

## Methods

### Training deep learning-based interatomic potential
The deep learning-based interatomic PE model for the Ga-F-Li-Cl chemical system was trained using the *Smooth Edition*[38] version 2.1.5 of DeepMD-kit package[17]. The *embedding* net and *fitting* net sizes were (24, 48, 96) and (240, 240, 240), respectively. The cutoff radius *rcut* was 9 Å and the smoothing parameter *rcut_smth* was set to 5.5 Å. The fitting network was trained by minimizing the loss function $L = p_eL_e+$

$p_fL_f$, where $L_e$ and $L_f$ were the loss in energy and force, respectively, and $p_e$ and $p_f$ were the corresponding prefactors varying from 0.02 to 1 and 1000 to 1, respectively. In the minimization of the loss function, the exponential learning rate decayed every *decay_steps* of 5000 from an initial value of 0.001. A total number of $8 × 10^5$ (Supplementary Fig. 2) iterations of training batches (minimization steps) were used to produce the final DeepMD PE model, until the root mean square error (RMSE) in energy and forces became constant, with the RMSE values for energy and forces being <1 meV/atom, and <70 meV/Å, respectively (see Supplementary Fig. 2). The DeepMD PE model was trained using atomic configurations obtained from the AIMD simulation frames of various stable structures in the Ga-F-Li-Cl quaternary chemical space and slab-like structures of $LiCl|GaF_3$ (where a multilayer slab of [001] LiCl and [001] $GaF_3$ is interfaced "|"), as illustrated in Supplementary Fig. 1 and listed in Supplementary Table 1. To generate atomic configurations for the training data, a melt and quench strategy was utilized. Each structure was heated in AIMD simulations at a constant rate from $T = 0\,K$ to the respective highest temperature as shown in Supplementary Table 1 for 10 ps in the NVT ensemble and a time step of 2 fs. Subsequently, the temperature was kept constant for 30 ps, after which the structure was cooled from the highest temperature to $T = 0\,K$ in 10 ps under a constant cooling rate. Additionally, to sample a larger configuration space and incorporate the effect of volume change, AIMD simulations were done at high temperatures as shown in Supplementary Table 1 in the NPT ensemble and a time step of 2 fs. In total, over 700k AIMD frames were generated, out of which 250k atomic configurations were randomly selected to train the DeepMD PE model. Among the 250k configurations, 80% of them were used for training and 20% for validation.

### DFT calculations
First-principles DFT calculations were performed to do structural relaxation and obtain the bulk modulus of the compounds investigated in this work. All calculations were performed using the Vienna Ab initio Simulation Package[39] (VASP) version 6.2. Ion-electron interactions were represented by projector-augmented wave potentials[40]. The generalized gradient approximation (GGA) parameterized by Perdew-Burke-Ernzerhof[41] (PBE) was used to account for the electronic exchange and correlation. The DFT-D2 method of Grimme was used to include the Van der Waals interaction. The wave functions were expanded in a plane-wave basis with an energy cutoff of 500 eV and a reciprocal-space discretization of at least 30 $k$-points per Å$^{-1}$ was used to sample the Brillouin Zone. The convergence criteria were set as $10^{-6}$ eV for electronic loops and 0.01 eV/Å for ionic loops. Ab initio molecular dynamics (AIMD) simulations were performed to generate atomic configurations for the training datasets as well as to obtain the tensile stress-strain response and Li-ion conductivities for benchmarking the trained DeepMD PE model. A gamma-point-only sampling of $k$-space, a plane-wave energy cutoff of 500 eV, and a time step of 2 fs were used. In all our simulations, large supercells with cell length of at least 10 Å in each direction were used. For such large supercells, one $k$-point is sufficient to get reasonable accuracy in the properties of the materials (Supplementary Fig. 14). NVT ensemble calculations were performed with the Nosé-Hoover thermostat, while the NPT ensemble calculations were done with the Langevin thermostat in non-spin mode.

### Classical molecular dynamics
LAMMPS[19] package was used to perform classical MD simulations using the trained DeepMD interatomic PE model of Ga-F-Li-Cl chemical space. All MD simulations were performed in either the NVT or NPT ensembles (as specified). The Nose-Hoover thermostat and a time step of 1 fs were used. The $T_{damp}$ and $P_{damp}$ parameter was set to 0.2 ps and 1 ps, respectively. In the constant-stress MD simulations (Fig. 1c), the fluctuations in the applied external shear stress $σ_{yz}$, accumulated shear

strain γ$_{yz}$, and total PE $E$ were removed by replacing each quantity's (σ$_{yz}$, γ$_{yz}$, and $E$) value at time $t$ by its average value over a time interval of $[t − \Delta t, t + \Delta t]$, where $\Delta t = 5$ ps. Fig. 1c shows the values after this averaging. The constant strain rate MD simulations at $T = 300$ K (Fig. 2a) were performed by deforming the simulation cell on the $yz$ plane under a constant strain rate of $10^9$ s$^{-1}$, while all other boundaries (except $yz$) were controlled using the NPT equations of motion to zero pressure.

## Data availability

The authors declare that the data supporting the findings of this study are available within the paper and its Supplementary information files. The raw data for Figs. 1–3 are provided in the Source Data file. Source data are provided with this paper.

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

## Acknowledgements

The authors acknowledge the NREL HPC resources for providing computing facilities. This research used 7BM of the National Synchrotron Light Source II, a U.S. Department of Energy (DOE) Office of Science User Facility operated for the DOE Office of Science by Brookhaven National Laboratory under Contract No. DE-SC0012704. This work was supported by the Assistant Secretary for Energy Efficiency and Renewable Energy, Vehicle Technologies Office, of the U.S. Department of Energy under Contract No. DE-AC02-05CH11231, under the Advanced Battery Materials Research (BMR) Program.

## Author contributions

S.G., X.Y. and G.C. conceived the project. S.G. performed all the computational simulations. X.Y. did all the experiments. All authors contributed to the writing of the paper.

## Competing interests

The authors declare no competing interests
