## [Peer Review File · Nature Communications]

REVIEWER COMMENTS

Reviewer #1 (Remarks to the Author):

In this manuscript, the authors have suggested criteria for the formation of clay-like materials from a mixture of rigid salts, using both computational and experimental analyses. The authors claim that molecular solids governed by Van der Waals bonding should be formed through a kinetically sluggish anion exchange process to create clay-like materials. This manuscript has clarified the microscopic features of clay-like properties, and the research has the potential to provide insights into designing clay-like superionic conductors. I would recommend publishing the manuscript in Nature Communications after addressing the following questions and comments:

(1) Given the claim of the authors, the formation of molecular solid is one of the criteria of clay-like materials formation. Authors only suggest the SbCl_3 , InI_3 , and GaI_3 as molecular solids. To strengthen the study, it is necessary to perform a systematic screening of the material database to suggest other potential molecular solid compounds that could exhibit clay-like properties.

(2) Regarding on the reference 10, can you discuss on the role of the addition of AlF_3 in clay-like material formation based on the criteria the authors suggested?

(3) Why NaCl-GaF_3 do not show clay-like properties? Is there an effect of alkali ion on clay-like material properties?

(4) Please discuss on the local structure around Li at STZ or SB.

(5) The manuscript would benefit from showing the Ga-K edge EXAFS data to at least 6.0 \AA , which would provide further insights into the material's properties.

Reviewer #2 (Remarks to the Author):

This paper proposes a mechanism for the peculiar case that when two powders are mixed, they usually become powders, but when $[\text{LiCl}+\text{GaF}_3]$ is mixed, they become soft-clay-like.

Although I am supportive of this study for the publication, following comments and suggestions need to be answered before considering its publication.

Major Comments:

1. The author suggested three criteria that must be met to form soft clay from ionic solids derived from the LiCl-GaF_3 solid mixture simulation. However, the generalization of such a claim without theoretical analysis of other types of ionic solid mixtures remains uncertain.

2. While ionic liquids exhibit frequent anion exchanges, ionic solids in powder form will show very few or no anion exchanges. In the case of soft clay, which seems to lie between ionic liquids and ionic solids, moderate anion exchanges are expected. The author is kindly requested to provide a detailed explanation of "(ii) the kinetics of the reaction must be slow enough to avoid complete anion exchange and phase separation" and how this can be validated without conducting simulations involving other types of ionic solid mixtures.

3. Considering that the DeepMD potential energy surface was trained primarily on crystalline structures, a pertinent question arises regarding its performance in the amorphous state. Addressing this concern necessitates a comprehensive evaluation, backed by evidence and thoughtful comparisons. By systematically exploring its predictive capabilities and rigorously assessing its performance against known amorphous systems, we can ascertain the extent to which the DeepMD potential energy surface effectively models amorphous materials.

4. One of the primary focuses of this article revolves around the transition from an ordered structure to an amorphous one. However, the temperatures employed for sampling the atomic

configurations may be relatively low and it potentially hinder the exploration of transition states within the configuration space. Given the importance of this phase transition, it would be advantageous to consider a wider range of temperatures to achieve a more comprehensive understanding of the system's behavior and transition mechanisms.

5. The training set and validation set are derived from the same AIMD trajectory. Considering the potential correlation between these sets, it is advisable to incorporate independently obtained trajectories to validate the trained model, ensuring a more robust and reliable assessment.

6. In the context of DFT calculations, I kindly request further elaboration on the rationale behind opting for gamma-point-only calculations and their adequacy for the study. Your insights would be invaluable in better comprehending the appropriateness and sufficiency of this approach.

Minor Comments

1. There is a typo in line 83: The unit is 'meV/A²,' but I think, it should be 'meV/A.'

Reviewer #3 (Remarks to the Author):

Inorganic Li-superionic conductors with a soft feature, synthesized by mixing rigid-salts, have been reported recently and show very interesting mechanical and electrochemical properties. The soft feature and high ionic conductivity make it especially attractive as ion conductor additives in cathode for solid-state batteries. This manuscript tries to combine theoretical calculation and experiment to elucidate the microscopic features responsible for soft clay-formation. It is useful for better understanding this system. However, I think the current version needs revision, because several key points need to be clarified.

Comments

1. Line 116 (Page3): Could you explain what is "GaCl₃-like units"?
2. Line 230-231 (Page 7): "Soft-clay will not form if the soft component is deficient or in excess". Please give a more detailed explanation on why excess soft component cannot form soft-clay.
3. From the theoretical view, how to design the nominal composition that can meet one of the criteria of "neither of the salt components is in much excess"?
4. As you claimed that "LiF and GaCl₃, does not lead to a soft-clay." I wonder why LiCl+GaCl₃ can form a soft-clay but LiF+GaCl₃ cannot. What is the reaction mechanism behind?
5. To propose possible design strategies, more instructional advices are interesting. For example, as LiCl-1Ga₂O₃ and 1Li₂O-1GaF₃ are unsuccessful, is it possible to get a soft clay from other LiCl-oxides or LiF-oxides design?
6. The composites are amorphous state, so the soft feature may be relevant to the supercooled liquid region. If yes, how to use this relation to design the soft clay? I think the authors should consider carefully and state this in the main text.
7. Considering that Ga is a rare metal, how about using other cation elements replacing/doping Ga? Does the clay-feature originate from Ga-based anion units ONLY?

RESPONSE TO REVIEWERS' COMMENTS

We thank the reviewers for carefully reading our manuscript and for their comments. Below are our answers, with changes to the main manuscript text and supplementary information (SI) highlighted in blue.

REVIEWER COMMENTS

Reviewer #1 (Remarks to the Author):

In this manuscript, the authors have suggested criteria for the formation of clay-like materials from a mixture of rigid salts, using both computational and experimental analyses. The authors claim that molecular solids governed by Van der Waals bonding should be formed through a kinetically sluggish anion exchange process to create clay-like materials. This manuscript has clarified the microscopic features of clay-like properties, and the research has the potential to provide insights into designing clay-like superionic conductors. I would recommend publishing the manuscript in Nature Communications after addressing the following questions and comments:

Response: We thank the reviewer for the positive comments and recommending publishing our work in Nature Communications.

(1) Given the claim of the authors, the formation of molecular solid is one of the criteria of clay-like materials formation. Authors only suggest the SbCl_3 , InI_3 , and GaI_3 as molecular solids. To strengthen the study, it is necessary to perform a systematic screening of the material database to suggest other potential molecular solid compounds that could exhibit clay-like properties.

Response: We thank the reviewer for this suggestion. We have now searched the Materials Project database and identified other molecular solids (MS) in addition to already suggested GaCl_3 , SbCl_3 , InI_3 and GaI_3 . We have also identified the possible anion exchange reaction with a maximum thermodynamic driving force to form the corresponding molecular solid. The data summarized in the table below (Supplementary Table 3) is now added to the Supplementary Information (SI).

page 9 (main text): We searched the Materials Project database for other molecular solids and the anion exchange reaction with the highest TDF to form them (see Supplementary Table 3), which has the potential to form soft-clay.

page 11 (SI):

Supplementary Table 3. Different possible molecular solid (MS) compounds and the corresponding mp-id obtained from the Materials Project (mp) database. The highlighted compounds are discussed in more detail in the main text. Possible anion exchange reaction with the most negative reaction energy E_{rxn} to form the MS is also listed. Li-based salts were considered as one of the reactants in the anion exchange reaction.

mp-id	MS Compounds	Reaction	Er _{rxn} (eV/atom)
mp-29831	TaCl ₅	0.455 LiCl + 0.545 TaF ₅ → 0.091 TaCl ₅ + 0.455 LiTaF ₆	-0.097
mp-28391	SiCl ₄	0.571 LiCl + 0.429 SiF ₄ → 0.286 Li ₂ SiF ₆ + 0.143 SiCl ₄	-0.036
mp-23250	UCl ₆	0.857 LiCl + 0.143 UF ₆ → 0.143 UCl ₆ + 0.857 LiF	-0.037
mp-22897	HgCl	0.5 LiCl + 0.5 HgF → 0.5 HgCl + 0.5 LiF	-0.23
mp-30086	GeCl ₄	0.429 GeF ₄ + 0.571 LiCl → 0.143 GeCl ₄ + 0.286 Li ₂ GeF ₆	-0.173
mp-23280	AsCl ₃	0.75 LiCl + 0.25 AsF ₃ → 0.25 AsCl ₃ + 0.75 LiF	-0.098
mp-23290	PtCl ₂	0.667 LiCl + 0.333 PtF ₂ → 0.333 PtCl ₂ + 0.667 LiF	-0.362
mp-22908	BiCl ₃	0.75 LiCl + 0.25 BiF ₃ → 0.25 BiCl ₃ + 0.75 LiF	-0.063
mp-570355	PbCl ₄	0.8 LiCl + 0.2 PbF ₄ → 0.2 PbCl ₄ + 0.8 LiF	-0.213
mp-30952	GaCl₃	0.6 LiCl + 0.4 GaF ₃ → 0.2 GaCl ₃ + 0.2 Li ₃ GaF ₆	-0.093
mp-22855	HgCl ₂	0.667 LiCl + 0.333 HgF ₂ → 0.333 HgCl ₂ + 0.667 LiF	-0.289
mp-22872	SbCl₃	0.75 LiCl + 0.25 SbF ₃ → 0.25 SbCl ₃ + 0.75 LiF	-0.074
mp-571518	WCl ₆	0.143 WF ₆ + 0.857 LiCl → 0.857 LiF + 0.143 WCl ₆	-0.122
mp-23176	SbCl ₅	0.455 LiCl + 0.545 SbF ₅ → 0.455 LiSbF ₆ + 0.091 SbCl ₅	-0.167
mp-23307	NbCl ₅	0.545 NbF ₅ + 0.455 LiCl → 0.455 LiNbF ₆ + 0.091 NbCl ₅	-0.088
mp-30092	TiCl ₄	0.571 LiCl + 0.429 TiF ₄ → 0.286 Li ₂ TiF ₆ + 0.143 TiCl ₄	-0.068
mp-29866	SnCl ₄	0.571 LiCl + 0.429 SnF ₄ → 0.286 Li ₂ SnF ₆ + 0.143 SnCl ₄	-0.171
mp-570005	SbBr ₃	0.75 LiBr + 0.25 SbF ₃ → 0.25 SbBr ₃ + 0.75 LiF	-0.126
mp-568846	TaBr ₅	0.455 LiBr + 0.545 TaF ₅ → 0.091 TaBr ₅ + 0.455 LiTaF ₆	-0.101
mp-570285	SiBr ₄	0.571 LiBr + 0.429 SiF ₄ → 0.286 Li ₂ SiF ₆ + 0.143 SiBr ₄	-0.036
mp-1208424	TeBr ₄	0.8 LiBr + 0.2 TeF ₄ → 0.2 TeBr ₄ + 0.8 LiF	-0.181
mp-23317	AsBr ₃	0.75 LiBr + 0.25 AsF ₃ → 0.25 AsBr ₃ + 0.75 LiF	-0.152
mp-23177	HgBr	0.5 LiBr + 0.5 HgF → 0.5 HgBr + 0.5 LiF	-0.301
mp-1207486	ZrBr ₄	0.571 LiBr + 0.429 ZrF ₄ → 0.286 Li ₂ ZrF ₆ + 0.143 ZrBr ₄	-0.021
mp-27642	PaBr ₅	0.833 LiBr + 0.167 PaF ₅ → 0.167 PaBr ₅ + 0.833 LiF	-1.54
mp-23292	HgBr ₂	0.667 LiBr + 0.333 HgF ₂ → 0.333 HgBr ₂ + 0.667 LiF	-0.385
mp-27399	SbBr ₃	0.75 LiBr + 0.25 SbF ₃ → 0.25 SbBr ₃ + 0.75 LiF	-0.126
mp-28601	NbBr ₅	0.545 NbF ₅ + 0.455 LiBr → 0.455 LiNbF ₆ + 0.091 NbBr ₅	-0.098
mp-30953	GaBr₃	0.4 GaF ₃ + 0.6 LiBr → 0.2 GaBr ₃ + 0.2 Li ₃ GaF ₆	-0.128
mp-569814	TiBr ₄	0.571 LiBr + 0.429 TiF ₄ → 0.286 Li ₂ TiF ₆ + 0.143 TiBr ₄	-0.081
mp-574086	SiBr ₄	0.571 LiBr + 0.429 SiF ₄ → 0.286 Li ₂ SiF ₆ + 0.143 SiBr ₄	-0.036
mp-23216	SnBr ₄	0.429 SnF ₄ + 0.571 LiBr → 0.286 Li ₂ SnF ₆ + 0.143 SnBr ₄	-0.209
mp-23288	AlBr ₃	no reaction	
mp-567604	GeBr ₄	0.429 GeF ₄ + 0.571 LiBr → 0.143 GeBr ₄ + 0.286 Li ₂ GeF ₆	-0.2
mp-1203015	TaBr ₅	0.455 LiBr + 0.545 TaF ₅ → 0.091 TaBr ₅ + 0.455 LiTaF ₆	-0.101
mp-30954	GaI₃	0.25 GaF ₃ + 0.75 LiI → 0.25 GaI ₃ + 0.75 LiF	-0.188
mp-1078195	SiI ₃	0.5 LiI + 0.5 SiF ₃ → 0.167 SiI ₃ + 0.25 Li ₂ SiF ₆ + 0.083 Si	-0.056
mp-23218	AsI ₃	0.75 LiI + 0.25 AsF ₃ → 0.25 AsI ₃ + 0.75 LiF	-0.248
mp-567789	InI₃	0.75 LiI + 0.25 InF ₃ → 0.25 InI ₃ + 0.75 LiF	-0.225
mp-22859	HgI	0.5 LiI + 0.5 HgF → 0.5 HgI + 0.5 LiF	-0.385
mp-23182	SnI ₄	0.2 SnF ₄ + 0.8 LiI → 0.2 SnI ₄ + 0.8 LiF	-0.291
mp-570884	Tel ₄	0.8 LiI + 0.2 TeF ₄ → 0.2 Tel ₄ + 0.8 LiF	-0.287
mp-29109	SiI ₃	no reaction	
mp-30930	AlI ₃	no reaction	
mp-635441	SiI ₄	0.571 LiI + 0.429 SiF ₄ → 0.286 Li ₂ SiF ₆ + 0.143 SiI ₄	-0.054
mp-23266	GeI ₄	0.2 GeF ₄ + 0.8 LiI → 0.2 GeI ₄ + 0.8 LiF	-0.257

(2) Regarding on the reference 10, can you discuss on the role of the addition of AlF₃ in clay-like material formation based on the criteria the authors suggested?

Response: In ref.10, materials with composition $2\text{LiCl}-x\text{AlF}_3-(1-x)\text{GaF}_3$ ($0.5 \leq x \leq 0.9$) were synthesized by ball-milling. It was found that clay-like mechanical behavior is observed only for composition $x \sim 0.8$. For $x > 0.8$ (AlF_3 -rich), a powder state remained, while for $x < 0.8$, a viscous gel-like state was obtained. The absence of soft-clay-like features in AlF_3 -rich systems indicates that GaF_3 is essential for obtaining soft-clay-like mechanical properties. AlF_3 does not have any thermodynamic driving force (TDF) to react with LiCl or GaF_3 (Fig. R1-1), and it does not satisfy our predicted *criteria (i) - the combination of salts must have a thermodynamic driving force for anion exchange leading to units that form molecular solids*. This is why AlF_3 -rich compositions do not form a soft-clay. However, for compositions $x < 0.8$, a viscous state was formed because addition of AlF_3 lowers the effective melting point of the salt mixtures [ref. 10], and the resulting system is a mixture of rigid solids dispersed in a liquid which can have viscous-like behavior [*Nat. Mater.* 10, 56–60 (2011); doi:10.1126/article.29483]. According to our criteria, AlF_3 does not have an *explicit* role in soft-clay formation, and the soft-clay-like behavior seen in ref. 10 is primarily due to the reaction between GaF_3 and LiCl . Addition of AlF_3 does change the effective melting point of the combined $\text{LiCl}-\text{GaF}_3-\text{AlF}_3$ mixture and local melting might occur during ball milling. This local melting can indirectly affect clay-formation by altering the kinetics of the anion exchange reaction between LiCl and GaF_3 and modulating the optimal ratio of $\text{LiCl}:\text{GaF}_3$ required for clay formation. A detailed investigation of how lowering of melting point alters the optimal ratio of $\text{LiCl}:\text{GaF}_3$ for soft-clay formation is beyond the scope of the present work.

Fig. R1-1. The reaction energy of $\text{LiCl}-\text{AlF}_3$ (left) and $\text{GaF}_3-\text{AlF}_3$ (right) reaction, taken from entries in the Materials Project database.

page 9 (main text): Additionally, in recent experiments¹⁰, ball milling a mixture of $2\text{LiCl}-x\text{AlF}_3-(1-x)\text{GaF}_3$ revealed soft-clay-like behavior when $x \sim 0.8$. For $x > 0.8$ (AlF_3 -rich), a powder state remained, while for $x < 0.8$, a viscous gel-like state was obtained. AlF_3 does not have any TDF to react with LiCl or GaF_3 . This is likely the reason why AlF_3 -rich compositions do not form a soft-clay, and why GaF_3 remains essential for obtaining soft-clay-like mechanical properties in the $2\text{LiCl}-x\text{AlF}_3-(1-x)\text{GaF}_3$ mixture. The authors found that for compositions $x < 0.8$, a viscous state was formed. We attribute this to the fact that addition of AlF_3 lowers the effective melting point of the salt mixtures,¹⁰ and the resulting system might be a mixture of rigid solids dispersed in a liquid with viscous-like behavior³.

(3) Why NaCl-GaF₃ do not show clay-like properties? Is there an effect of alkali ion on clay-like material properties?

Response: The chemical reaction between $\text{NaCl}+\text{GaF}_3$ does not have any thermodynamic driving force to form molecular solid GaCl_3 . The reaction energy of $3\text{NaCl} + \text{GaF}_3 \rightarrow 3\text{NaF} + \text{GaCl}_3$ is positive $E_{\text{rxn}} = 0.474$ eV (obtained from the Materials Project reaction energy calculator). No molecular solid is expected to form

when NaCl and GaF₃ react, and thus we believe that it will not show clay-like properties, as also seen in prior experiments [*United States Patent US-20210376378-A1 (2021)*]. We do not expect any explicit effect of alkali ions on clay formation. Clay will form if there is a thermodynamic driving force for molecular solid formation by anion exchange reaction between the two precursors and the kinetics of such a reaction is sluggish. We have added the NaCl-GaF₃ reaction to the SI.

page 11 (main text): Until now only Li-based systems have shown pliable behavior, however, we believe that these strategies are not limited to Li-based systems and can be applied to discover other beyond-Li soft clay-like systems if mixtures with appropriate reaction energy can be found.

page 10 (SI): $3\text{NaCl} + \text{GaF}_3 \rightarrow 3\text{NaF} + \text{GaCl}_3$ $E_{\text{rxn}} = 0.474 \text{ eV (46 kJ mol}^{-1}\text{)}$

(4) Please discuss on the local structure around Li at STZ or SB.

Response: We appreciate the reviewer's comment. We have now discussed the local structure around Li at STZs and SBs. Supplementary Fig. 15 (below) shows the $g(r)$ around Li atoms with the highest and lowest D^2_{min} values. The $g(r)$ was averaged over the strain interval $\gamma_{\text{yz}} = 0.09 - 0.14$. We found that Li atoms in the areas involved in plastic deformation have a Cl-rich environment, as evidenced by the lower Li-F peak for Li atoms with $D^2_{\text{min}} > 90\%$ of maximum compared to those with D^2_{min} values $< 10\%$ of maximum (Supplementary Fig. 15, below). We have added the figure and the discussion in the main text and SI.

page 6 (main text): ...(see Supplementary Fig. 15 for a similar plot of $g(r)$ around Li atoms).

page 16 (SI):

Supplementary Fig. 15 The Li-Cl and Li-F pair distribution function $g(r)$ of Li-atoms with largest ($D^2_{\text{min}} > 90\%$ of maximum) and smallest ($D^2_{\text{min}} < 10\%$ of maximum) non-affine displacement values. The $g(r)$ values were averaged over the strain interval $\gamma_{\text{yz}} = 0.09 - 0.14$. The Li atoms in the areas involved in plastic deformation have a Cl-rich environment, as evidenced by the lower Li-F peak for Li atoms with $D^2_{\text{min}} > 90\%$ of maximum compared to those with D^2_{min} values $< 10\%$ of maximum.

(5) The manuscript would benefit from showing the Ga-K edge EXAFS data to at least 6.0 Å, which would provide further insights into the material's properties.

Response: We thank the reviewer for the suggestion. We have included the Ga-K edge EXAFS data to the Supplementary Information (Supplementary Fig. 13, below) since in the main text we focus on comparing the first coordination peak. The weak magnitude of Fourier transformed EXAFS at higher radial

distance indicates a highly disordered Ga environment, which is consistent with the amorphous nature of the clay-like samples.

page 7 (main text): The weak magnitude of Fourier transformed EXAFS at higher radial distance indicates a highly disordered Ga environment, which supports the amorphous nature of the soft clay-like samples (see Supplementary Fig. 13 for the EXAFS plotted to a radial distance of 6 Å).

page 15 (SI):

Supplementary Fig. 13. Ga K-edge EXAFS of different materials, bulk GaF_3 , bulk GaCl_3 , and clay-like 2LiCl-1GaF_3 and 3LiCl-1GaF_3 . $|\chi(r)|$, magnitude of Fourier transformed EXAFS.

Reviewer #2 (Remarks to the Author):

This paper proposes a mechanism for the peculiar case that when two powders are mixed, they usually become powders, but when [LiCl+GaF₃] is mixed, they become soft-clay-like.

Although I am supportive of this study for the publication, following comments and suggestions need to be answered before considering its publication.

Response: We thank the reviewer for the positive feedback.

Major Comments:

1. The author suggested three criteria that must be met to form soft clay from ionic solids derived from the LiCl-GaF₃ solid mixture simulation. However, the generalization of such a claim without theoretical analysis of other types of ionic solid mixtures remains uncertain.

Response: We appreciate the reviewer's concern. The three criteria for soft clay formation referred to in our work are based on results derived from *both computational simulations and experiments*. Our computational analysis on LiCl+GaF₃ solid mixture suggests that the formation of molecular solid-like (MS) units is key to soft-clay-like mechanical properties. MS-like units act as sites for shear transformation zones (STZs), which being inherently soft can get activated at low stress and lead to soft plasticity. Solid mixtures where there is no thermodynamic driving force for molecular solid formation will not form soft clay (criterion #1). This criterion #1 is supported by several pieces of experimental evidence as reported in our work as well in prior work. As an example, NaCl+GaF₃ does not form a soft clay (as reported in prior work [*United States Patent US-20210376378-A1 (2021)*]), which we can explain because there is no thermodynamic driving force for molecular solid formation. The reaction energy of $3\text{NaCl} + \text{GaF}_3 \rightarrow 3\text{NaF} + \text{GaCl}_3$ is positive $E_{\text{rxn}} = 0.474$ eV (obtained from the Materials Project reaction energy calculator).

Regarding criterion #2, "the kinetics of the reaction must be slow enough to avoid complete anion exchange and phase separation". This criterion is proposed based on our experimental results on 3LiCl-SbF₃, 3LiI-GaF₃, and 3LiI-InF₃, which did not form a soft-clay on ball-milling. We find that anion exchange happens in all three cases. However, the anion exchanged product phase separates into macroscopic phases. In the non-clay case of 3LiI-1InBr₃, XRD peaks corresponding to LiBr and InI₃ (anion exchanged products) were seen after just 20 minutes of ball milling (Supplementary Fig. 10d), indicating a rapid anion exchange and phase separation. Conversely, for the soft clay-like 3LiCl-1GaF₃, no signature of anion exchange completion or phase separation was seen even after 24 hours of ball milling. Additionally, ball-milling the compounds that would constitute the terminal products of anion-exchange, LiF and GaCl₃, does not lead to a soft-clay. This indicates that sluggish kinetics of the anion exchange reaction is important for soft-clay formation. A complete anion exchange and phase separation will not lead to soft-clay formation.

Regarding criterion #3, the ratio of salts must be in an appropriate range. This criterion is backed by ours and prior experimental reports on the xLiCl-GaF₃ system, where soft-clay is not formed for $x < 2$.

The three criteria that we suggest are backed by computational simulations and experiments and appear general. Of course, we cannot exclude that more information will refine these criteria, as is the process of science. We do not expect that more calculations on the systems considered in our study will change the current conclusions. To improve the clarity of our claims, we have moved the statements on the three criteria from results to the discussion section.

page 8 (main text): ~~Hence, to form soft clay from ionic solids, three criteria must be met—(i) the combination of salts must have a thermodynamic driving force for anion exchange leading to units that form molecular solids, (ii) the kinetics of the reaction must be slow enough to avoid complete anion exchange and phase separation, and (iii) the ratio of salts must be in an appropriate range, ensuring that neither of the salt components is in much excess.~~

page 10 (main text): Our combined computational simulations and experimental results suggest that to form soft-clay from ionic solids, three criteria must be met - (i) the combination of salts must have a thermodynamic driving force for anion exchange leading to units that form molecular solids, (ii) the kinetics of the reaction must be slow enough to avoid complete anion exchange and phase separation, and (iii) the ratio of salts must be in an appropriate range, ensuring that neither of the salt components is in much excess.

page 10 (main text): ... our combined computational simulations and experimental results suggest that ...

2. While ionic liquids exhibit frequent anion exchanges, ionic solids in powder form will show very few or no anion exchanges. In the case of soft clay, which seems to lie between ionic liquids and ionic solids, moderate anion exchanges are expected. The author is kindly requested to provide a detailed explanation of "(ii) the kinetics of the reaction must be slow enough to avoid complete anion exchange and phase separation" and how this can be validated without conducting simulations involving other types of ionic solid mixtures.

Response: While we have no quantitative information about the anion exchange rate, we agree with the reviewer that it is likely that the extent of anion exchange in soft-clay systems tends to lie between ionic liquids and ionic solids. Therefore, regulating the kinetics of the anion exchange is important for soft-clay formation. To achieve soft clay-like mechanical behavior, it is necessary to have both soft and hard components and them being interfaced with each other [Meunier, *A. Clays. (Springer Science & Business Media, 2005)*]. If the diffusion of the cation or anion species is too fast, then soft and hard components will phase segregate into macroscopic separate phases and soft clay-like deformation will not be achieved. In our case, anion exchange occurs when the rigid salts (e.g., LiCl + GaF₃) react, which leads to the formation of molecular solid (MS) units (GaCl₃), which acts as the soft components. If the kinetics of this anion exchange reaction is fast, such that it leads to completion of anion exchange and phase segregation of the molecular species into macroscopic phases, then the MS-like soft units will not be interfaced between the hard solids, and soft clay-like mechanical response will not occur. We indeed find that ball-milling the compounds that would constitute the terminal products of anion-exchange, LiF and GaCl₃, does not lead to a soft-clay. This validates our claim that "*the kinetics of the reaction must be slow enough to avoid complete anion exchange and phase separation*". To form a soft-clay, anion exchange must occur, but the products should not phase segregate into macroscopic separate phases.

We have also validated this claim with experiments on other salt mixtures 3LiCl-SbF₃, 3LiI-GaF₃, and 3LiI-InF₃. These salt mixtures did not form a soft-clay on ball-milling. We find that anion exchange happens in all three cases. However, the anion exchanged product phase separates into macroscopic phases. In the non-clay case 3LiI-1InBr₃, XRD peaks corresponding to LiBr and InI₃ (anion exchanged products) were seen after just 20 minutes of ball milling (Supplementary Fig. 10d), indicating a rapid anion exchange and phase separation, while for the soft clay-like 3LiCl-1GaF₃, no signatures of anion exchange completion or phase separation were seen even after 24 hours of ball milling. This further validates our claim that a complete anion exchange and phase separation will not lead to soft-clay formation. As we already mention in the manuscript, in the cases which did not form a soft-clay during ball milling, temperature can be regulated to suppress the kinetics of the anion exchange reaction. We have now

added additional discussion in the revised manuscript. We certainly agree that it may be possible to refine this theory further, but at this point we have no experimental or computational information to do so.

page 9 (main text): The kinetics of anion exchange is also critical for soft-clay formation. If the anion exchange reaction's kinetics is fast, such that it leads to anion exchange completion and phase segregation of the molecular species into macroscopic phases, then the MS-like soft units will not be interfaced between the hard solids, and soft clay-like mechanical response will not occur.

page 9 (main text): Suppressing the kinetics of anion exchange in such non-clay forming cases may be possible if the temperature during ball milling can be regulated.

3. Considering that the DeepMD potential energy surface was trained primarily on crystalline structures, a pertinent question arises regarding its performance in the amorphous state. Addressing this concern necessitates a comprehensive evaluation, backed by evidence and thoughtful comparisons. By systematically exploring its predictive capabilities and rigorously assessing its performance against known amorphous systems, we can ascertain the extent to which the DeepMD potential energy surface effectively models amorphous materials.

Response: We appreciate the reviewer's comment. We have now tested the DeepMD potentials accuracy on properties of several amorphous systems in the Ga-F-Li-Cl chemical space. We created amorphous phases of different chemical systems LiCl, LiF, GaCl₃, LiGaCl₄, and Li₃GaF₆ by melting and quenching each of the crystalline phases using AIMD-DFT and DP (DeepMD-LAMMPS) model. Melting was obtained by performing NVT simulation for each of the structures at $T = 3000$ K for $t = 5$ ps. The atomic configuration at $t = 5$ ps was subsequently relaxed at $T = 0$ K to obtain the amorphous phase. We compare the radial pair distribution function of the amorphous systems, the energy of the amorphous systems with respect to their crystalline phase, and the density of the amorphous systems obtained with DFT and the trained DP model. Supplementary Fig. 5 (below) shows such a comparison. We find that the trained DP model accurately predicts the properties of the amorphous structures. We have added these results in the supplementary information. We thank the reviewer for raising this point, which further ascertains the accuracy of the trained DP model.

page 3 (main text): We also examined our trained PE model's accuracy in describing the properties of several amorphous systems in the Ga-F-Li-Cl chemical space and found good agreement between model's prediction and actual DFT values (see Supplementary Note 1.3 and Supplementary Fig. 5).

page 7 (SI): **Supplementary Note 1.3: Testing the accuracy of DeepMD PE model on properties of amorphous systems**

The trained DeepMD PE model was further tested on the properties of different amorphous systems in the Ga-F-Li-Cl chemical space. Amorphous phases of different chemical systems LiCl, LiF, GaCl₃, LiGaCl₄, and Li₃GaF₆ were created by melting and quenching each of the crystalline phases using AIMD-DFT and DeepMD-LAMMPS (DP). Melting was obtained by performing NVT simulation for each of the structures at $T = 3000$ K for $t = 5$ ps. The atomic configuration at $t = 5$ ps was subsequently relaxed at $T = 0$ K to obtain the amorphous phase. Supplementary Fig. 5 shows a comparison of the radial pair distribution function of the amorphous systems, the energy of the amorphous systems with respect to their crystalline phase, and the density of the amorphous systems obtained with DFT and the trained DP model. The results obtained with DFT and the trained DP model are in good agreement.

Supplementary Fig. 5: (a) Comparison of the radial pair distribution function $g(r)$ of amorphous (amr.) LiF and amorphous Li_3GaF_6 obtained with AIMD-DFT and trained DP model. (b) The relative energy of the amorphous (amr.) phase of different chemical systems with respect to their crystalline (cry.) phase obtained with AIMD-DFT and trained DP model. (c) Comparison of the density of different amorphous systems obtained with AIMD-DFT and with the trained DP model.

4. One of the primary focuses of this article revolves around the transition from an ordered structure to an amorphous one. However, the temperatures employed for sampling the atomic configurations may be relatively low and it potentially hinder the exploration of transition states within the configuration space. Given the importance of this phase transition, it would be advantageous to consider a wider range of temperatures to achieve a more comprehensive understanding of the system's behavior and transition mechanisms.

Response: We appreciate the reviewer's suggestion. We have now done simulations at different temperatures. In addition to the earlier heating simulation at $T = 900$ K, we have now done similar simulations at $T = 1200$ K and $T = 1500$ K. Supplementary Fig. 7a (below) shows the element-wise radial pair distribution function $g(r)$ before and after the high temperature ($T = 900$ K, 1200 K, and 1500 K) MD simulation of $t = 50$ ps. We find that in all the three temperature cases, the Ga-F and Li-Cl peak decrease, while the Ga-Cl and Li-F peak increase, indicating that anion exchange occurs when the two salts are mixed. This anion exchange is consistent with the thermodynamic driving force and with the experimental EXAFS (Fig. 3a, main text). The Ga-F and Li-Cl peak decreases by larger amounts on increasing the temperature from 900 K - 1500 K, indicating that the extent of the anion exchange increases with increasing the temperature. However, anion exchange behavior is still captured by simulations at $T = 900$ K, which is consistent with experimental EXAFS (Fig. 3a, main text). We also simulated the shear stress-strain response of the systems obtained after heating MD runs at three different temperatures. A procedure similar to that described in the main text for strain-controlled MD simulation was used to obtain the stress-strain response. Supplementary Fig. 7b (below) shows the shear stress-strain response of the structures at an applied shear strain rate of 10^9 s $^{-1}$ on the yz plane. The stress-strain response curves of the 3 different structures look very similar, indicating that they show similar mechanical behavior. We have now added these new results in the supplementary information. We believe that $T = 900$ K (as used earlier) is sufficient to capture the anion exchange behavior and the mechanical response of the amorphous system.

page 3 (main text): ... (see Supplementary Note 2.1 for the system's behavior at other temperatures) ...

page 9 (SI): **Supplementary Note 2.1: Effect of heating temperature on the behavior of the amorphous system**

In addition to the heating simulation at $T = 900$ K (Supplementary Fig. 6), we also did similar simulations at $T = 1200$ K and $T = 1500$ K. Supplementary Fig. 7a shows the element-wise radial pair distribution function $g(r)$ before and after the high temperature ($T = 900$ K, 1200 K, and 1500 K) MD simulation of $t = 50$ ps. We find that in all the three temperature cases, the Ga-F and Li-Cl peak decrease, while the Ga-Cl and Li-F peak increase, indicating that anion exchange occurs when the two salts are mixed. This anion exchange is consistent with the thermodynamic driving force and with the experimental EXAFS (Fig. 3a, main text). The Ga-F and Li-Cl peak decreases by larger amounts on increasing the temperature from 900 K - 1500 K, indicating that the extent of the anion exchange increases with increasing the temperature. However, anion exchange behavior is still captured by simulations at $T = 900$ K, which is consistent with experimental EXAFS (Fig. 3a, main text). We also simulated the shear stress-strain response of the systems obtained after heating MD runs at three different temperatures. A procedure similar to that described in the main text for strain-controlled MD simulation was used to obtain the stress-strain response. Supplementary Fig. 7b shows the shear stress-strain response of the structures at an applied shear strain rate of 10^9 s $^{-1}$ on the yz plane. The stress-strain response curve of the 3 different structures looks very similar, indicating that they show similar mechanical behavior. We believe that $T = 900$ K is sufficient to capture the reaction mechanism and mechanical behavior of the amorphous system.

Supplementary Fig. 7: (a) The element-wise radial pair distribution function $g(r)$ plotted at $t = 1$ ps and $t = 50$ ps of the MD simulation at $T = 900$ K, $T = 1200$ K, and $T = 1500$ K. (b) Shear stress (σ_{yz}) strain (γ_{yz}) response of the amorphous structures at $T = 300$ K at a constant applied shear strain rate of 10^9 s $^{-1}$. The

three curves correspond to the behavior of the three systems obtained after heating MD runs at $T = 900$ K, $T = 1200$ K, and $T = 1500$ K.

5. The training set and validation set are derived from the same AIMD trajectory. Considering the potential correlation between these sets, it is advisable to incorporate independently obtained trajectories to validate the trained model, ensuring a more robust and reliable assessment.

Response: We agree with the reviewer that even though training and validation data sets were selected randomly, there might be correlation between the training and validation data sets as they are derived from the same AIMD trajectory. This is why, to further test our model, we generated independent trajectories by doing additional AIMD simulations. We tested our model on those trajectories that were not included in the training and validation set, and were obtained independently of them. As we show in Supplementary Fig. 3, our trained model accurately predicts the energy and forces on atoms in those trajectories. Additionally, we have now assessed the model's accuracy by calculating properties of different amorphous systems in the Ga-F-Li-Cl chemical space (see response to your comment #3) and further demonstrated the reliability of our trained model.

6. In the context of DFT calculations, I kindly request further elaboration on the rationale behind opting for gamma-point-only calculations and their adequacy for the study. Your insights would be invaluable in better comprehending the appropriateness and sufficiency of this approach.

Response: We appreciate the reviewers' concern. In all our simulations, we have used large supercells with cell length of at least 10 Å in each direction. This is a common approach for large supercells. For such large supercells, one k -point is sufficient to get reasonable accuracy in the properties of the materials as the volume of the Brillouin zone of a cell scales inversely with its real-space volume. Below we show the change in energy/atom of the different chemical systems as a function of k -grid size. The simulation cell size in all the systems was at least 10 Å in each direction. We find that the change in energy on going from one k -point to 18 k -points is only ~1 meV/atom, which is small. The computational cost significantly decreases with using just one k -point and we see that there is no significant gain in the total energy's accuracy with using more k -points. Hence, we have used just one- k point in our calculations. We have added this figure and the discussion in the revised manuscript.

page 12 (main text): In all our simulations, large supercells with cell length of at least 10 Å in each direction were used. For such large supercells, one k -point is sufficient to get reasonable accuracy in the properties of the materials (Supplementary Fig. 14)

page 16 (SI):

Supplementary Fig. 14: The energy of the different chemical systems obtained using DFT calculations with different k -grid sizes. $N_{k\text{-point}}$ is the total number of k -points in the k -grid. The energy is referenced to the value obtained with one k -point.

Minor Comments

1. There is a typo in line 83: The unit is 'meV/Å²,' but I think, it should be 'meV/Å.'

Response: We thank the reviewer for pointing this out. The typo has been corrected in the revised manuscript.

page 3 (main text): ... 70 meV/Å²

Reviewer #3 (Remarks to the Author):

Inorganic Li-superionic conductors with a soft feature, synthesized by mixing rigid-salts, have been reported recently and show very interesting mechanical and electrochemical properties. The soft feature and high ionic conductivity make it especially attractive as ion conductor additives in cathode for solid-state batteries. This manuscript tries to combine theoretical calculation and experiment to elucidate the microscopic features responsible for soft clay-formation. It is useful for better understanding this system. However, I think the current version needs revision, because several key points need to be clarified.

Response: We thank the reviewer for the comments. We have now addressed them all in the revised manuscript.

Comments

1. Line 116 (Page3): Could you explain what is "GaCl₃-like units"?

Response: GaCl₃-like units are molecular complexes which have a 4-fold Ga-Cl tetrahedral coordination similar to that in bulk GaCl₃. The exact structure of GaCl₃-like units is not the same as molecular complexes in bulk GaCl₃, hence we have used the word "like". We have clarified this in the revised manuscript.

page 4 (main text): ...GaCl₃-like units, which are molecular complexes that have a 4-fold Ga-Cl tetrahedral coordination similar to that in bulk GaCl₃ (Fig. 3b).

2. Line 230-231 (Page 7): "Soft-clay will not form if the soft component is deficient or in excess". Please give a more detailed explanation on why excess soft component cannot form soft-clay.

Response: To achieve a soft-clay-like property, the soft and hard units must be interfaced with each other. If the soft units phase segregates into a macroscopic phase then soft-clay will not form. Hence, if there is an excess of the soft components, the soft-units can phase segregate into a macroscopic phase, which is detrimental for achieving a soft-clay like property. This is analogous to the appropriate ratio of water and pyrophyllite-like minerals to form natural soft-clay. If water is in excess, then soft-clay will not form. We have added additional discussion to explain this in the revised manuscript.

page 7 (main text): If the soft component is in excess, soft-units can phase segregate into a macroscopic phase, which is detrimental for achieving a soft-clay like property.

3. From the theoretical view, how to design the nominal composition that can meet one of the criteria of "neither of the salt components is in much excess"

Response: This is a challenging question. Our computational simulation results indicate the importance of anion exchange reaction and molecular solid units formation for realizing a soft-clay-like behavior. On the one hand a large enough driving force is needed for the ion exchange so that enough molecular units form. But on the other hand we don't want a too large driving force that would cause a large-scale phase separation. In addition, the particle size of the precursors and the ball-milling conditions may affect the kinetic conditions and the nominal composition. At this point we don't have a precise way to predict the exact mixing ratio that will result in soft-clay formation. Developing a theoretical model to predict the nominal composition for soft-clay formation is beyond the scope of this work and warrants further investigation.

4. As you claimed that "LiF and GaCl₃, does not lead to a soft-clay." I wonder why LiCl+GaCl₃ can form a soft-clay but LiF+GaCl₃ cannot. What is the reaction mechanism behind?

Response: Respectfully, we do not agree with the reviewer's comment that "LiCl+GaCl₃ can form a soft-clay". As has been reported in prior works [*ACS Energy Lett.* **6**, 2006–2015 (2021)], LiCl+GaCl₃ does not form a soft-clay. Also we do not make any claims in the paper that LiCl+GaCl₃ can form a soft-clay. Both LiCl+GaCl₃ and LiF+GaCl₃ do not form a soft-clay because GaCl₃ (molecular solid) is already phase separated into macroscopic phases in both these cases. However, LiCl+GaF₃ can form a soft-clay due to reasons discussed in detail in the manuscript.

5. To propose possible design strategies, more instructional advices are interesting. For example, as LiCl-1Ga₂O₃ and 1Li₂O-1GaF₃ are unsuccessful, is it possible to get a soft clay from other LiCl-oxides or LiF-oxides design?

Response: We thank the reviewer for the helpful suggestions. However, we screened the Materials Project database and found that LiCl-oxides combinations either have no thermodynamic driving force (TDF) for anion change or do not have a maximum TDF for forming the targeted molecular solids (MS), both of which fails to satisfy our first criterion for soft-clay formation. The reaction energy between LiCl and oxides is summarized in the table below:

Targeted MS	Reaction	E_{rxn} (eV/atom)	Formation of MS?
BiCl ₃	0.545 LiCl + 0.455 Bi ₂ O ₃ → 0.273 LiBi ₃ (ClO ₂) ₂ + 0.091 Li ₃ BiO ₃	-0.037	No
SbCl ₃	No anion exchange reaction between LiCl and Sb ₂ O ₃		No
SnCl ₄	No anion exchange reaction between LiCl and SnO ₂		No
TaCl ₅	No anion exchange reaction between LiCl and Ta ₂ O ₅		No
NbCl ₅	No anion exchange reaction between LiCl and Nb ₂ O ₅		No
WCl ₆	No anion exchange reaction between LiCl and WO ₃		No

Additionally, since there are no F-based molecular solids (more details are provided in response to comment "1" of reviewer #1), the mixture of LiF with oxides is not expected to show soft-clay-like properties. The anion exchange reaction between LiF and oxides will not form any molecular solids and it does not satisfy our criterion #1 for clay formation. To design other possible systems, we have now identified other Cl, Br, and I-based molecular solids (MS) in addition to already suggested GaCl₃, SbCl₃, InI₃, and GaI₃. The possible anion exchange reaction with a maximum thermodynamic driving force to form the corresponding molecular solid is also listed (see our response to comment "1" of reviewer #1).

6. The composites are amorphous state, so the soft feature may be relevant to the supercooled liquid region. If yes, how to use this relation to design the soft clay? I think the authors should consider carefully and state this in the main text.

Response: We agree with the reviewer that the soft-clay systems are in an amorphous state and show a glass-transition at a temperature lower than room temperature. This suggests that at room temperature there might be regions in the amorphous system that are in a supercooled liquid state [*J. Phys. Chem.* 1996, 100, 31, 13200–13212; *Nature* 410, 259–267 (2001)]. Additionally, the soft molecular units could be those regions in a supercooled liquid state, however, we do not have enough evidence to make a detailed claim in the paper. We do believe that there is a connection between molecular solids that can form glass and can remain in a supercooled liquid state at room temperature, and soft-clay formation. However, identifying which systems can form a glass or not is an active area of research [*Nature* 410, 259–267 (2001); *Phys. Rev. Lett.* **91**, 115505 (2003); *Nature Materials* 21, 165–172 (2022)] and it has not been established whether every substance can be put into a glass form. We have stated the connection between soft molecular units and them being in supercooled liquid state in the revised manuscript. We appreciate the reviewer's insightful comment.

page 10 (main text): The soft clay-systems examined in this study are in an amorphous state and exhibit a glass-transition⁹ at $T_g \sim -60$ °C. This implies that at room temperature, there might be regions in the amorphous system that are in a supercooled liquid state^{36,37} and those regions may contain molecular units that are critical to the system's soft mechanical response. Finding molecular solids capable of forming glass and remaining in a supercooled liquid state at room temperature, as well as developing methods to synthesize such molecular solid based compounds, can aid in the design of other soft-clay systems and requires further exploration.

7. Considering that Ga is a rare metal, how about using other cation elements replacing/doping Ga? Does the clay-feature originate from Ga-based anion units ONLY?

Response: We thank the reviewer for the suggestion. We tried to replace Ga with other metals that can also form molecular solids after anion exchange reaction. We tried the combinations of LiCl-SbF₃ and LiI-InBr₃, since SbCl₃ and InI₃ are both identified as molecular solids. However, the products of these two reactions remain in the powder state without the desirable soft-clay feature. From the XRD, we found that in these two cases, the anion exchanged product phase segregated into macroscopic phases, thereby not satisfying our *criterion #2*.

On the other hand, the doping strategy has proved successful in a previous literature report [*Adv. Sci.* **9**, 2204633 (2022)]. Materials with composition 2LiCl-xAlF₃-(1-x)GaF₃ ($0.5 \leq x \leq 0.9$) were synthesized by ball-milling. It was found that clay-like mechanical behavior is observed only for composition $x \sim 0.8$ and GaF₃ is essential to achieve clay-like mechanical softness. For $x > 0.8$ (AlF₃-rich), a powder state remained, while for $x < 0.8$, a viscous gel-like state was obtained. Therefore, completely replacing Ga with other cation elements remains challenging, since the three criteria have to be satisfied at the same time. It appears that the inherent chemistry of Ga-F-Cl chemical space helps satisfy all the three criteria for soft-clay formation, however, the microscopic reasons leading to the special behavior by Ga-F-Cl chemical systems warrants further investigation and could be a topic of a separate study. It is viable to dope other metal cations into the GaF₃-based composition to reduce the cost. However, such an effort warrants further investigations and is beyond the scope of the current work.

page 9 (main text): Additionally, in recent experiments¹⁰, ball milling the mixture of 2LiCl-xAlF₃-(1-x)GaF₃ revealed soft-clay-like behavior when $x \sim 0.8$.

page 10 (main text): Until now, only the LiCl-GaF_3 , and $\text{LiCl-AlF}_3\text{-GaF}_3$ systems have demonstrated soft-clay-like properties. The underlying chemistry of Ga-F-Cl chemical space appears to assist in satisfying all the three criteria for soft-clay formation, however, the microscopic reasons that distinguish Ga-F-Cl chemical systems warrants further investigation.

REVIEWERS' COMMENTS

Reviewer #1 (Remarks to the Author):

The authors have earnestly addressed all the issues I raised in the original manuscript. Therefore, I recommend its publication in Nature Communications

Reviewer #2 (Remarks to the Author):

For the most part, the authors responded well to the comments I raised. I do have one additional question about the question #3 and its response.

It is interesting that only the 5 ps long molecular dynamics simulations from each methodology agree well with each other. Regarding question 3, it would be better to show a parity plot of the forces obtained from the DFT calculations and the DeePMD PE to show the performance of the trained model. Also, the bin size of the radial distribution function (Supplementary Figure 5(a)) seems to be a bit large; it might be better to use a smaller bin size.

Reviewer #3 (Remarks to the Author):

Thanks for the kind responses. I still have a simple question. As you mentioned that "anion exchange takes place during the ball milling of a LiCl-GaF₃ mixture forming GaCl₃-like molecular units". Is it possible to prepare the GaCl₃-like molecular units by melting and fast-quenching method?

RESPONSE TO REVIEWERS' COMMENTS

We thank the reviewers for their comments. Below are our answers, with changes to the supplementary information (SI) highlighted in **blue**.

REVIEWERS' COMMENTS

Reviewer #1 (Remarks to the Author):

The authors have earnestly addressed all the issues I raised in the original manuscript. Therefore, I recommend its publication in Nature Communications

Response: We thank the reviewer for recommending publishing our work in Nature Communications.

Reviewer #2 (Remarks to the Author):

For the most part, the authors responded well to the comments I raised. I do have one additional question about the question #3 and its response.

It is interesting that only the 5 ps long molecular dynamics simulations from each methodology agree well with each other. Regarding question 3, it would be better to show a parity plot of the forces obtained from the DFT calculations and the DeePMD PE to show the performance of the trained model. Also, the bin size of the radial distribution function(Supplementary Figure 5(a)) seems to be a bit large; it might be better to use a smaller bin size.

Response: We appreciate the reviewer's comment. We have now included a parity plot comparing the atomic forces in the amorphous structures obtained with AIMD-DFT simulations at $T = 3000$ K and predicted by the DeepMD-PE model (see Supplementary Fig. 5d-f, below). Good agreement between the trained model's predictions and the DFT values is obtained. We have also re-plotted Supplementary Fig. 5a with a larger bin size.

page 8 (SI):

Supplementary Fig. 5: (a) Comparison of the radial pair distribution function $g(r)$ of amorphous (amr.) LiF and amorphous Li_3GaF_6 obtained with AIMD-DFT and trained DP model. (b) The relative energy of the amorphous (amr.) phase of different chemical systems with respect to their crystalline (cry.) phase obtained with AIMD-DFT and trained DP model. (c) Comparison of the density of different amorphous systems obtained with AIMD-DFT and the trained DP model. (d - f) Parity plot comparing the atomic forces obtained by AIMD-DFT simulations at $T = 3000$ K and predicted by DP model for the different amorphous systems.

Reviewer #3 (Remarks to the Author):

Thanks for the kind responses. I still have a simple question. As you mentioned that "anion exchange takes place during the ball milling of a LiCl-GaF₃ mixture forming GaCl₃-like molecular units". Is it possible to prepare the GaCl₃-like molecular units by melting and fast-quenching method?

Response: The melt-and-quench method is indeed useful for obtaining amorphous materials. However, in our specific case, the formation of soft-clay should satisfy the criteria that the kinetics of the reaction must be slow enough to avoid complete anion exchange and phase separation. The melting points of LiCl and GaF₃ are 605 °C and 800 °C, respectively, whereas the boiling point of GaCl₃ is 200.5 °C. Hence, it is likely that during the melting of a LiCl and GaF₃ mixture, the GaCl₃ formed would evaporate.